# Determination of Dapoxetine Hydrochloride in Human Plasma by HPLC–MS/MS and Its Application in a Bioequivalence Study

**DOI:** 10.3390/molecules27092707

**Published:** 2022-04-22

**Authors:** Xin Zhang, Zhanwang Gao, Fei Qin, Kehan Chen, Jiansong Wang, Lingli Wang

**Affiliations:** 1Guangzhou Higher Education Mega Center, School of Pharmaceutical Sciences, Guangzhou University of Chinese Medicine, No. 232, Waihuandong Road, Guangzhou 510006, China; 20202110169@stu.gzucm.edu.cn (X.Z.); 20201110890@stu.gzucm.edu.cn (Z.G.); 20211121047@stu.gzucm.edu.cn (K.C.); 2Baiyunshan Pharmaceutical General Factory, Guangzhou Baiyunshan Pharmaceutical Holdings Co., Ltd., Guangzhou 510515, China; 102536@byszc.com

**Keywords:** dapoxetine, method validation, HPLC–MS/MS, pharmacokinetic studies

## Abstract

Dapoxetine is used for the treatment of premature ejaculation. The present study developed an HPLC–MS/MS method to determine the levels of dapoxetine in human plasma processed using simple protein precipitation. Dapoxetine-d_7_ was selected as the internal standard. The established method was performed using a mass spectrometer equipped with an electrospray ionization source in multiple positive ion reactions to monitor the mode using the precursor-to-product ion transitions of *m/z* 306.2–157.2 and *m/z* 313.2–164.2 for dapoxetine-d7 and dapoxetine, respectively. The method was evaluated based on its selectivity, linearity, limit of quantification, precision, accuracy, matrix effects, dilution integrity, stability, and extraction recovery. As a result of the model used in the present study, the validated linear ranges of dapoxetine were determined to be 2.00~1000 ng/mL in plasma, and the selectivity, precision, accuracy, dilution integrity, stability, and extraction recovery met the accepted standard. No matrix interference was observed. The method was successfully validated and applied to pharmacokinetic studies in healthy Chinese volunteers during the fasting and postprandial periods, respectively.

## 1. Introduction

Premature ejaculation (PE), which is characterized by ejaculation within about 1 min during sexual experiences, refers to a type of male sexual dysfunction [1]. Dapoxetine, a selective serotonin reuptake inhibitor (SSRI), is a short-acting SSR1 and has been used extensively for the treatment of PE [2]. As the only SSR1 approved for PE treatment, dapoxetine can block the presynaptic membrane 5-hydroxytryptamine (5-HT) transporter, increasing the level of 5-HT in the synaptic cleft and activating the postsynaptic 5-HT2C and 5-HT1A receptors, thereby increasing ejaculation time [3].

The safety and efficacy of dapoxetine have been reported. Clinical evidence has indicated that 30 mg and 60 mg of on-demand dapoxetine significantly improves intravaginal ejaculatory latency time (IELT) [4,5]. Compared to alternative forms of care (such as topical desensitizing creams), dapoxetine also shows longer mean IELT times [6,7]. Only a few studies have proposed determination methods for dapoxetine. The existing analytical methods for dapoxetine evaluation generally lack any evaluation of matrix effects [8], require long chromatographic times [9], and lack sensitivity [10].

In the present study, a selective and sensitive HPLE/MS-MS method was established and validated to determine the concentration of dapoxetine in human plasma. The plasma samples were processed by means of simple protein precipitation, and dapoxetine-d_7_ was selected as the internal standard (IS) due to its physicochemical properties and mass spectral signal response, which are similar to those of dapoxetine. The linear range of our method was determined to be 2.00~1000 ng/mL, a marked improvement in sensitivity over previously reported methods, allowing it to be used in our bioequivalence studies of dapoxetine in the fasting and postprandial states. In addition, the method developed in the present study was also utilized to evaluate dapoxetine bioequivalence studies in the fasting and postprandial states.

## 2. Results and Discussion

### 2.1. HPLC–MS/MS Analysis of Dapoxetine

Our analysis of plasma dapoxetine levels features simple protein precipitation followed by LC-MS/MS. The overall chromatographic analysis time was 2.8 min. The corresponding proposed fragment assignments are shown in Figure 1.

### 2.2. Method Validation

#### 2.2.1. System Applicability

The relative standard deviation (RSD) of the analyte, internal standard retention time, and peak area ratio were ≤0.4%, 0.4% and 8.1%, respectively. The blank substrates from six different sources had no obvious interference with the analyte and internal standard and did not impair quantitative analysis.

#### 2.2.2. Linearity and Sensitivity

The calibration curves were created by plotting the peak area ratios of the various analytes to internal standards versus the nominal concentration of the analyte standards. In our calibration curves, the correlation coefficient R^2^ 0.99 and all of the standard calibration curves showed good linearity within the range using least squares regression analysis. The LLOQ, representing the lowest concentration on the calibration curves, was 2.0 ng/mL for dapoxetine. Correlation coefficients (r) over 0.99 and with an RSD of 0.04% proved that all of the calibration curves had an excellent linear relationship (Appendix A).

#### 2.2.3. Selectivity

In this assay, no matrix effect or interference were observed when comparing the results obtained from the blank substrates (six different sources), blank reagents, and blank matrices. Furthermore, no interference was observed between the analyte and the internal standard (Figure 1). In addition, the deviation in the accuracy of the LLOQ QC prepared from six matrices from different sources was less than 20%.

#### 2.2.4. Interference of Isotope Internal Standard with the Analyte

No interference was observed in the isotope internal standard working solution during analyte detection, and the solution did not affect the quantitative analysis, regardless of whether it was stored at room temperature for 41 h or at 4 °C for 42 d.

#### 2.2.5. Matrix Effect

The proximity of the matrix effects from nine different sources (fasting plasma, postprandial plasma, and hemolytic plasma) to the analyte and the internal target did not affect the quantitative analysis of the analyte (Table 1).

#### 2.2.6. Accuracy and Precision

The intra-batch and inter-batch precision and accuracy of the analytical method were determined by testing different concentrations of independent QC samples in plasma. As shown by the specific results in Table 2, the RSD meets the acceptance criteria.

#### 2.2.7. Extraction Recovery

After pre-treatment, the mean dapoxetine extraction recovery in plasma at the three QC levels was 101.0% (HC), 104.2% (MC), and 99.7% (LC), with a 2.3% RSD, while that of hyperlipoidemia was 100.8%, 103.4% and 98.9%, with a 2.2% RSD (Table 3).

#### 2.2.8. Dilution Reliability

Similarly, the dilution integrity test showed that QC samples could be diluted by up to a factor of five while retaining acceptable precision and accuracy (Table 4).

#### 2.2.9. Stability

The samples were stable in plasma at room temperature for up to 41 h, and if refrigerated, they were stable for up to 36 d at 4 °C (Table 5).

### 2.3. Pharmacokinetic Study

The validated HPLC–MS/MS methods were successfully applied to evaluate the pharmacokinetic profiles of DP in healthy Chinese volunteers during the fasting and postprandial periods. The main pharmacokinetic parameters are summarized in Table 6. The maximum plasma concentrations (Cmax) were 448.9 ± 203.57 ng/mL and 549.1 ± 201.70 ng/mL for the fasting and postprandial periods, respectively. The elimination half-life (t1/2) was 17.681 ± 6.0084 h and 17.858 ± 5.9342 h for the fasting and postprandial periods, respectively. These results showed that the effects of food resulted in no significant differences in the pharmacokinetic parameters. A single dose of 60 mg DP demonstrated a good safety profile.

## 3. Materials and Methods

### 3.1. Chemical and Reagents

Dapoxetine HCl (DP) (99.8% purity) and its deuterated analogue DP-d7 (99.8% purity) were obtained from TLC Pharmaceutical Standards (Shanghai, China). Chromatography-grade methanol and acetonitrile were sourced from Thermo Fischer Scientific (Shanghai, China). Chromatography-grade methylic acid was obtained from Tianjin Comio Chemical Reagent Limited Company (Tianjin, China); ultrapure water was prepared using an ELGA LabWater device, and drug-free (blank)human plasma and blood were obtained from the People’s Hospital of Weifang High-tech Industrial Development Zone (the anticoagulant used was EDTA-K2).

### 3.2. Instruments and Conditions

HPLC–MS/MS was performed using Shimadzu HPLC systems and an AB Sciex Mass analyzer (TRIPLE QUAD 4500). The HPLC systems included a high-performance liquid chromatography pump (LC-20ADXR), an automatic sampler (SIL-30ACMP), and a column temperature box (CTO-20AC). Chromatographic separation was carried out on an LC-20ADXR (Shimadzu, Japan) using an HPLC column Ultimate XB C18 (4.6 × 50 mm, 5 µm). Mobile phase A was composed of an aqueous 0.1% formic acid solution. Mobile phase B comprised an acetonitrile solution containing 0.1% formic acid. The flow rate was set to 0.800 mL/min. The automatic sampler temperature was set at 4 °C. Initial Pump B Conc was set at 30%. The injection volume was 5.00 ul, and the column temperature was set at 35 °C. The following gradient program was used for sample separation: 0–1.2 min, 50% B; 1.2–1.3 min, 50–95% B; 1.3-2.0 min, 95% B; 2.0–2.1 min, 95–30% B; 2.1–2.8 min, 30% B. Electrospray ionization (ESI)was used as the ion source, and a positive pattern was chosen for the ionization mode. The multi-reaction monitoring model was used in detection mode, and the ion spray voltage was set at 5500 V. The turbo ion spray temperature was set at 550 °C, the curtain gas type was set at 30.0 psi, and the CAD gas type was 9. The nebulizing gas, gas1, was set at 50.0 psi; gas 2, the auxiliary gas, was set at 55.0 psi. Both gases were nitrogen; the entrance voltage was set at 10.0 V, the acquisition time was 2.80 min. The optimized mass spectrometric parameters are shown in Table 7.

### 3.3. Preparation of Calibration Standards and Quality Controls

The DP (1.00 mg/mL) and IS (1.00 mg/mL) stock solutions were prepared in methanol. The working solutions for the standard curves and quality control (QC) samples were obtained by serially diluting the stock solution with methanol/water (50:50, *v/v*). All of the prepared working solutions were stored in glass bottles and stored in the refrigerator at 4 °C.

For the standard curve, the DP stock solutions were diluted to the subsequent working solutions (0.04, 0.08, 0.2, 1, 3, 10.0, 18, 20 μg/mL) and spiked with blank plasma at the concentrations of 2, 4, 10, 50, 150, 500, 900, and 1000 ng/mL.

For quality control, the IS stock solutions were diluted to the subsequent working solutions (0.04, 0.12, 1.2, 15, 80 μg/mL) and spiked with blank plasma at the concentrations of 2 ng/mL for the lower limit of quantification QC (LLOQ QC), 6 ng/mL for the low QC (LQC), 60 ng/mL for the medium QC (MQC), 750 ng/mL for the high QC (HQC), and 4000 ng/mL for the dilution QC (DQC).

### 3.4. Sample Preparation

First, 50 μL of analyte and 25 μL of methanol/water (50:50, *v*/*v*) were added to 50 μL of blank plasma and vortexed for 5 min. After centrifuging (1700× *g* for 15 min, 4 °C), 50.0 μL of supernatant was transferred into a 96-well plate containing 450 μL methanol/water (50:50, *v*/*v*) and vortexed for 5 min. The treated samples were injected into the HPLC–MS/MS system for analysis. Zero and blank samples, with and without IS, respectively, were included for each set of standards.

### 3.5. Data Analysis

All of the data were processed using Analyst 1.6.3, and the results were subjected to regression analysis using the weighted (W = 1/x^2^) least square method, in which the concentration of analyte was used as the abscissa, and the peak area ratio of the analyte to the IS was used as the ordinate.

### 3.6. Method Validation

#### 3.6.1. Calibration Curve

Eight standard calibration samples were prepared according to the method described in the “Section 3.3”, and then the standard calibration samples were treated according to the method described in the “Section 3.4”. The analysis results showed linear regression with the peak area ratio of the analyte to the internal standard as the ordinate and 1/x^2^ as the weight factor. The standard curve of the analyte in plasma was obtained, and the concentration of the corrected standard sample was back calculated according to the standard curve and the peak area ratio between the analyte and internal standard. The deviation in the accuracy of the back-calculated concentration and labeled concentrations of the standard calibration samples were required to be within ±15% (20% for LLOQ). At least 75% of each standard curve and at least six calibration samples met the above standards, and the R^2^ of the standard curve was not less than 0.98.

#### 3.6.2. Selectivity

The selectivity referred to:(1)The six blank matrix samples from different sources, blank matrix samples from mixed sources, and blank reagent samples were selected. Acceptance criteria: The minimum accuracy deviation in the quantitation quality control samples obtained from at least five blank substrates from different sources did not exceed ±20%; the interference peak response at the analyte retention time in the blank matrix of at least five different sources was lower than 20% of the analyte response in the lower quantitation limit obtained from the blank matrix from the same source. The interference peak response at the internal standard retention time was less than 5% of the internal standard response in the lower limit quality control samples obtained from blank matrix samples from the same source. The interference peak response of the blank matrix samples and blank reagent samples from mixed sources were lower than 20% of the response of the analyte in the lower quantification limit. The interference peak response at the internal standard retention time should be less than 5% of the internal standard response in the zero-concentration sample.(2)The control zero samples (CTL-0) and the upper limit of quantitation without IS (ULOQ-NO IS) samples that were treated to evaluate the interference between the analyte and the internal standard. Acceptance criteria: the interference peak response of the CTL-0 samples at the retention time of the analyte was lower than 20% of the response of the analyte in the lower limit of quantification; the interference peak response of the ULOQ-NO IS sample at an internal standard retention time IS had to be less than 5% of the internal standard response of the sample at the zero concentration.(3)The short-term storage of the isotope internal standard solution at room temperature and interference with the quantitative analysis of the analytes after long-term storage. Acceptance criteria: the interference peak response of the zero-concentration sample at the retention time of the analyte was less than 20% of the response of the analyte in the lower limit of quantification.

#### 3.6.3. Determination of Precision and Accuracy

LLOQ QC, LQC, MQC, and HQC4 concentration quality control samples were prepared. The intra- and inter-batch accuracy deviation and precision of each concentration quality control sample were calculated (six parallel samples for each batch, three batches), and the accuracy and precision of the analysis method were evaluated. Acceptance criteria: the average accuracy deviation within and between batches of each concentration control sample did not exceed ±15% (20% LLOQ QC), the RSD did not exceed 15% (20% for LLOQ QC), and there were at least five effective values for each concentration.

#### 3.6.4. Carryover

After the highest calibration standard, three blank samples were injected. Acceptable carryover was defined as the detector response for MQC was ≤20% of the LLOQ. For the IS, it was ≤5% of the average for the IS throughout the run.

#### 3.6.5. Matrix Effects

Nine blank plasma samples from different sources, i.e., three from fasting blood (normal plasma), three from postprandial blood (hyperlipaemia plasma), and three from cell cracking the hemolysis matrix of different donors, were added to the HQC, MQC, LQC, and internal standards. The matrix effect evaluation samples and pure solution control samples that were of the same concentration without matrix extract were prepared to evaluate the matrix effect (including hyperlipemia) on the accurate quantitative analysis of analytes by calculating the internal standard normalized matrix factors in each sample. Acceptance criteria: the relative standard deviation of the internal standard normalized matrix factors in the nine matrices from different sources could be no more than 15%.

#### 3.6.6. Lower Limit of Quantification and Linearity

The LLOQ QC and six parallel samples were prepared for each quality control sample concentration for three precision and accuracy evaluation analysis batches. Based on the precision and accuracy of the LLOQ QC, the rationale of the sample concentration setting and the determination reliability were evaluated. In addition, the sensitivity of the analytical method was determined by measuring the signal-to-noise ratio (S/N); that is, the ratio of the instrumental response of the analyte in the sample at the lower limit of quantification and the instrumental response observed at the retention time of the analyte in the blank matrix sample. The average deviation in the accuracy of the LLOQ QC did not exceed ±20%, and the RSD did not exceed 20%. As for sensitivity, the S/N of the analyte was less than 5.

#### 3.6.7. Dilution Integrity

The DQC was prepared for the dilution reliability evaluation, and the sample was diluted with blank plasma up to five times. Then, the sample was processed with six parallel samples. The accuracy of the deviation and precision of the different concentrations of quality control samples were obtained, and the reliability of the analytical method was confirmed when the analyte concentration in the determination sample exceeded the upper quantitative limit by no more than 4000 ng/mL. Acceptance criteria: The average deviation of the accuracy of the quality control sample did not exceed ±15%, the RSD of precision was less than 15%, and each concentration had at least five effective values.

#### 3.6.8. Stability

The stability must be evaluated in each step prior to the determination of the clinical samples to simulate the various storage and analysis conditions of the clinical samples as much as possible. These stability conditions mainly include the sample collection stability, the auto-sampler stability, and the repeated injection reproducibility of the pretreated plasma samples, as well as the short-term/long-term stability and five freeze–thaw cycles at different temperatures. Moreover, the stability of the stock solution and working solutions at room temperature or under freezing conditions (−20 °C and −80 °C) were also determined.

#### 3.6.9. Extraction Recovery

Extraction recoveries were assessed by comparing the responses of the samples after extraction from the corresponding quality control samples. The mean values of the six replicates for each sample were used for the assessment.

### 3.7. Pharmacokinetic Study

The validated method was applied to a DP bioequivalence study in healthy Chinese people under both fasting and fed condition. This bioequivalence study was designed as an open, single-center, single-dose, two-cycle, two-sequence, randomized, crossover trial. This study protocol was approved by the Ethics Committee of The Affiliated Hospital of Guizhou Medical University (Guizhou, China, the approval number is 2019035), and all of the subjects signed informed consent before the study. A total of 95 healthy subjects who received a single dose of 60 mg DP were efficiently analyzed over the course of two weeks. A safety evaluation was also performed in the present study. The clinical trial was also registered in the China Drug Trial database (http://www.chinadrugtrials.org.cn, accessed on 6 April 2019) with approval number CTR20191062.

## 4. Conclusions

This study established a simple and sensitive HPLC–MS/MS method to determine DP in human plasma. The method was successfully applied to a pharmacokinetic study using healthy Chinese volunteers. It will provide baseline and safety data for future clinical studies on the use of dapoxetine HCl.

## Figures and Tables

**Figure 1 molecules-27-02707-f001:**
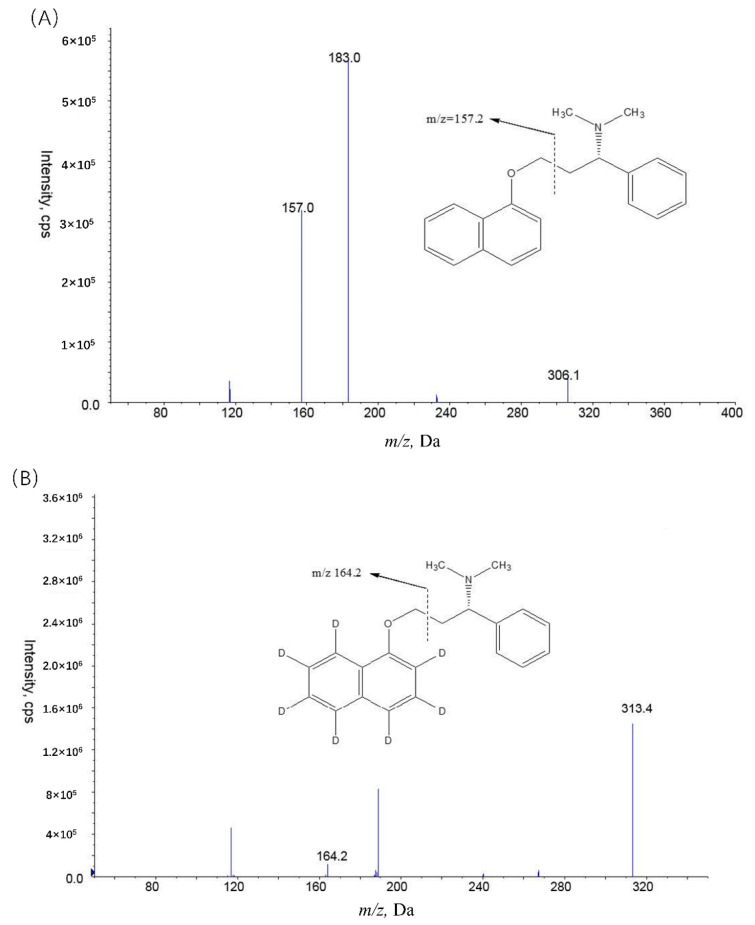
MS/MS spectra of DP (**A**) and IS (**B**).

**Table 1 molecules-27-02707-t001:** The matrix effect results of the different analytes.

Analyte N = 9	Internal Standard Normalized Matrix Factor		
	Mean Value	SD	RSD (%)
HQC	0.990 *	0.0135	1.4
	0.988 **		
	0.984 ***		
MQC	1.010 *	0.0200	2.0
	0.998 **		
	1.012 ***		
LQC	0.977 *	0.0373	3.8
	0.982 **		
	0.981 ***		

*** Fasting plasma; ** postprandial plasma; *** hemolytic plasma.

**Table 2 molecules-27-02707-t002:** Evaluation results for the accuracy and precision for intra-batch and inter-batch sample detection.

Analyte N = 6,6,6	Intra-Batch Mean	Intra-Batch RSD (%)	Deviation of Average Accuracy Intra-Batch (%)	Inter-Batch Mean N = 18	Inter-Batch RSD (%) N = 18	Average Accuracy Deviation Inter-Batches (%)N = 18
LLOQ QC	2.04	2.80	2.0	2.11	5.5	5.2
	2.14	5.90	7.20			
	2.13	6.40	6.60			
LQC	5.98	3.0	−0.30	6.13	3.4	2.2
	6.22	3.4	3.70			
	6.20	2.7	3.30			
MQC	60.1	3.4	0.10	61.9	3.8	3.2
	63.0	2.9	5.00			
	62.7	3.4	4.50			
HQC	752	1.8	0.30	753	2.4	0.4
	741	1.7	−1.20			
	765	2.7	2.00			

**Table 3 molecules-27-02707-t003:** The extraction recovery results.

Extraction Recovery	Normal Plasma			Hyperlipoidemia		
	HQC	MQC	LQC	HQC	MQC	LQC
The mean of analyte extraction recovery (%)	101.0	104.2	99.7	100.8	103.4	98.9
The relative standard deviation of analyte extraction recovery(%RSD)	2.3			2.2		

**Table 4 molecules-27-02707-t004:** Evaluation results of dilution reliability of plasma samples diluted up to five times.

Analyte N = 6	Mean Value (ng/mL)	SD	RSD (%)	Average Accuracy (%)	Mean Accuracy Deviation (%)
DQC (diluted by a factor of 5) 4000 ng/mL	4040	174	4.3	101.0	1.0

**Table 5 molecules-27-02707-t005:** Evaluation results of the stability of the analytical solution during short-term placement at room temperature.

Survey Sample Type N = 6	Duration of Investigation	Mean Value (ng/mL)	RSD (%)	Storage Condition
Stability control sample 1.00 (mg/mL)	0 h	0.935	3.9	RT
Stability test sample 1.00 (mg/mL)	41 h	0.920	2.3	RT
Stability control sample 1.00 (mg/mL)	0d	0.947	1.0	4 °C
Stability test sample 1.00 (mg/mL)	36d	1.04	1.3	4 °C

**Table 6 molecules-27-02707-t006:** The pharmacokinetic parameters of DP (mean ± SD) in healthy Chinese volunteers.

	Fasting	Post-Prandial
Tmax (h)	1.500 (0.5, 4)	1.750 (0.75, 4)
Cmax (ng/mL)	448.9 ± 203.57	549.1 ± 201.70
AUC0-t (h×ng/mL)	2671.599 ± 2298.5138	3196.242 ± 1210.5110
AUC0-∞ (h×ng/mL)	2810.615 ± 2387.0430	3383.230 ± 1416.7882
λz（1/h）	0.044 ± 0.0175	0.043 ± 0.0125
t1/2 (h)	17.681 ± 6.0084	17.858 ± 5.9342

**Table 7 molecules-27-02707-t007:** The optimized mass spectrometric parameters.

Compound Name	Multi-Reaction Monitoring (MRM)	Dwell Time (ms) *	Declustering Power (DP) (volts)	Collision Energy (CE) (volts)	Chromatographic Retention Time (min)
Dapoxetine	306.2–157.2	180	60.0	33.0	0.950
Dapoxetine-d7	313.2–164.2	180	60.0	34.0	0.950

* Note that here, residence time refers to the time spent scanning one ion pair at a time when the ion pairs were being monitored.

## Data Availability

The data presented in this study are available from the corresponding author upon request.

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
