# Peer review of "Determination of Dapoxetine Hydrochloride in Human Plasma by HPLC–MS/MS and Its Application in a Bioequivalence Study"

_molecules, 2022, doi:10.3390/molecules27092707_

Round 1

Reviewer 1 Report

First of all I suggest correction by native speaker. Manuscript is very hard to understand due to improper language used. Many words are with capital letters for some unknown reason. Wrongly use tense like present perfect instead of past simple or present simple. Manuscript should be rewritten to gain scientific soundness. At this point is not suitable for publication.

Comments:

- there is no such thing as “mass spectrum response” l45

- the matrix effects could not be similar to dapoxetine l46, in fact response is mitigated by IS

- I would disagree with such high linearity 2-1000 ng/mL. If so, please present the RSD of each point of calibration curve calculated by its equation

- Figure 1 is not readable

- what is good linearity? l69

- LLOQ, LOQ and LOD should be calculated not accepted as the lowest concentration l70

- according to selectivity, if matrix has no interference with analyte which is wrongly assumed, why to use IS? L75 and following

- I have serious doubts about the retention time of analytes. What is the dead time of this column and what is the retention factor if tR of analyty/IS is 0.95 min. Please provide chromatogram and explain how analytes are not influenced by dead volume/dead time

Reviewer 2 Report

This paper describes the development and validation of an analytical method using LC-MS/MS to quantify plasma levels of dapoxetine in a pharmacokinetic study as part of a clinical trial.  The authors follow standard procedures to demonstrate the validity of their method, including the sample preparation, recovery, sample stability, selectivity, matrix effects, and assay precision and accuracy.

At times, sections of this paper read like an instruction manual, setting the acceptance limits and using “should” and “shall” and “must”.  However, what is required is for the authors to tell us what was achieved.  I have noted the places where some rewriting is required below.

Minor revisions

Line

17 and 21 . . .  positive ion multiple reaction monitoring . .

20  Use lower case “s” for stability.  This is an error that recurs – please do a global correction

23  . . . healthy Chinese volunteers . .

36 . . improved . .

37  I don’t understand “care/non-dapoxetine”.  Do you mean to compare the use of dapoxetine to a placebo? Please clarify.

39-41  delete “And as we know” – we do not know.  Better to say: “The existing dapoxetine analytical methods generally lack any evaluation of matrix effects, require long chromatographic times and lack sensitivity. “

46-50  The linear range of our method is 2.00-1000ng/mL, a marked improvement in sensitivity over previously reported methods [refs] and allowing its use in our bioequivalence studies of dapoxetine in the fasting and post-prandial state.

53-54  Our analysis of plasma dapoxetine levels features a simple protein precipitation followed by LC-MS/MS.

Fig 1 This figure needs improved clarity – inconsistent line thickness, font size is too small and the structures are too faint.  In the legend identify the ion transitions used in the assay

68  In our calibration curves, the correlation coefficient R ³0.99, 

68 and 72  R is stated to be  ³0.99 and  ³0.999, respectively – which is it? Include the calibration curves in the Supplementary Data along with a chromatogram.

75-78  No matrix effect and no interference in the assay was observed when comparing the results from blank substrates (6 different sources) , blank reagents and blank matrices. Furthermore, no interference between the analyte and the internal standard were observed (Fig 1).

Table 3.  Please define terms HC, MC and LC.

98-99  The dilution integrity test showed that QC samples could be diluted by up to a factor of 5 while retaining acceptable precision and accuracy (Table 4).

101  . . .diluted by a factor of 5.

103  Samples . . . up to 41 h and if refrigerated, for up to . .

109  . . .in healthy Chinese volunteers . .

120-124  Dapoxetine HCl (DP) (99.8% purity) and its deuterated analogue DP-d7 (99.8% purity) were obtained from TLC Pharmaceutical Standards (City, Country). Chromatography grade methanol and acetonitrile were sourced from ThermoFischer Scientific (City, Country). Formic acid was sourced from Tianjin . . .

129-133  LC-MS/MS was performed with a Shimadzu HPLC system (LC-20ADXR pump, SIL-30ACMP autosampler, CTO-20AC column heater) fitted with an Ultimate XB C18 column (4.6 x 50 mm, 5mm) interfaced to a SCIEX Quad 4500 mass spectrometer.

134 and 135  The abbreviations MPA and MPB are not required.

137-145  Odd use of capitalisations – only required for proper nouns and the start of sentences.  Please identify the different gasses – UHP nitrogen? He for collision gas?

160-161 Please define abbreviations LLOQ QC, LQC, MQC, HQC and DQC.

165  delete “another” and replace with “a”.

183-186  I don’t understand the use of the expressions; “ . . shall not exceed . . ”, “ . . should not exceed . .” and “ . . . shall meet . . ”.  These expressions are appropriate in a set of instructions when setting the standards that must be met for a valid assay but this is description of the method and its limitations.  We want to know what you managed to achieve in your experiments.  Please re-write.

192, 194, 196, 199, 201, 220, 221, 246, 255, 258  ditto as above.  Please rewrite to tell us what was achieved.

203 Define CTL-0, ULOQ-NO IS,

207  Define uLOQ-NO IS

209 – 212  We need to know what was achieved, not the acceptance criteria.  Are you saying a loss of 20% is acceptable?

215 – 221  Please define abbreviations

231, 232 The abbreviations MER and MEP do not accord with the preceeding words – are they necessary?

242 The standard abbreviation for signal to noise ratio is S/N

283 - 285  . . . and was successfully applied to a pharmacokinetics study using healthy Chinese volunteers.  It will provide the baseline and safety data required for future clinical studies on the use of dapoxetine HCl.

Round 2

Reviewer 1 Report

Authors managed to improve their manuscript.